

# PATACSDB—the database of polyA translational attenuators in coding sequences

Malgorzata Habich[1], Sergej Djuranovic[2] and Pawel Szczesny[1,3]

[1] Institute of Biochemistry and Biophysics Polish Academy of Sciences, Department of Bioinformatics, Warsaw, Poland
[2] Department of Cell Biology and Physiology, Washington University School of Medicine, Saint Louis, MO, United States of America
[3] Faculty of Biology, Institute of Experimental Plant Biology and Biotechnology, University of Warsaw, Warsaw, Poland

Corresponding author
Pawel Szczesny,
szczesny@ibb.waw.pl

## ABSTRACT

Recent additions to the repertoire of gene expression regulatory mechanisms are polyadenylate (polyA) tracks encoding for poly-lysine runs in protein sequences. Such tracks stall the translation apparatus and induce frameshifting independently of the effects of charged nascent poly-lysine sequence on the ribosome exit channel. As such, they substantially influence the stability of mRNA and the amount of protein produced from a given transcript. Single base changes in these regions are enough to exert a measurable response on both protein and mRNA abundance; this makes each of these sequences a potentially interesting case study for the effects of synonymous mutation, gene dosage balance and natural frameshifting. Here we present PATACSDB, a resource that contain a comprehensive list of polyA tracks from over 250 eukaryotic genomes. Our data is based on the Ensembl genomic database of coding sequences and filtered with algorithm of 12A-1 which selects sequences of polyA tracks with a minimal length of 12 A's allowing for one mismatched base. The PATACSDB database is accessible at: http://sysbio.ibb.waw.pl/patacsdb. The source code is available at http://github.com/habich/PATACSDB, and it includes the scripts with which the database can be recreated.

## BACKGROUND

The classical view of the genetic information flow inside living cells—the transcription from DNA to RNA and finally translation of mRNA into protein—is a subject of continous modification for both direction of the flow and the number of players involved. After decades of research we keep accumulating evidences of several control points at different levels of these processes. The past studies were focused on transcriptional regulation, but more recently the regulation of gene expression at the level of translation drew researchers' attention. Translational regulation generally controls the amount of protein synthesised from a given mRNA through several mechanisms, targeting recruitment of ribosomes to

the transcript, elongation speed, termination and as a proxy to all these processes mRNA stability. Ribosome stalling—the pausing of ribosome during translational cycle—is recognized by components of several mRNA surveillance pathways. As a result of the impeded rate of ribosome along the mRNA, the transcript is endonucleolytically cleaved and nascent albeit incomplete protein product is degraded by proteasome (*Shoemaker & Green, 2012*). Over the years, we have got to know that certain sequence features can trigger ribosome stalling. These are damaged bases (*Cruz-Vera et al., 2004*), stable stem-loop structures (*Doma & Parker, 2006*), rare codons (*Letzring, Dean & Grayhack, 2010*), mRNAs lacking stop codons (so called non-stop mRNAs) (*Dimitrova et al., 2009*), runs of codons that encode consecutive basic aminoacids (*Kuroha et al., 2010*; *Brandman et al., 2012*), or finally, runs of adenines encoding poly-lysine tracks (*Koutmou et al., 2015*; *Arthur et al., 2015*).

We have recently shown that polyA tracks trigger a response in a different manner than runs of basic aminoacids (*Arthur et al., 2015*). In addition to stalling, they occasionally lead to ribosome sliding on mRNA transcript which results in production of additional frameshifted product next to the known and well annotated gene protein product. As such polyA track sequences may support programed translational frameshifts in such mRNA transcripts giving rise to alternative protein products from those genes. This feature of polyA track genes resembles programmed frameshifting observed in viral genes with slippery sequences however without a need for additional mRNA structures that induces ribosome stalling in known viral transcripts (*Chen et al., 2014*; *Yan et al., 2015*). The ultimate control over the production and stability of alternative transcripts from polyA track genes in Eukaryotes would be based on mRNA surveillance mechanisms, mainly non-sense mediated mRNA decay (NMD) or if the kinetic stall persists by no-go mRNA decay (NGD). PolyA tracks are highly conserved in genes among Eukaryotes and it is likely that they represent a universal translational attenuators or programed translational frameshift signals. Intrinsically this novel RNA motif plays an important role in balancing gene dosage and homeostasis of cellular environment. The level of attenuation, frameshifting and exact role of polyA tracks in organisms homeostasis is still to be elucidated.

## PATACSDB server

While there are several resources devoted to polyadenylation signals in genomic sequences, these have different sequence signature and refer to the processing of mRNA, not translation. No genomic database reports polyA tracks in coding sequences, therefore we have designed PATACSDB (PolyA Translational Attenuators in Coding Sequences DataBase), a resource devoted to collection of such features among eukaryotic organisms. In concordance with our experimental data from the controlled expression of reporter sequences or natural gene expression profiles we have designed a 12A-1 pattern, that is pattern of twelve adenines in coding region allowing for one mismatch. Based on our experiments, this is a minimal pattern that should result in reduction of expression by roughly 30%, a magnitude that can potentially have a measurable biological impact in human cells (*Arthur et al., 2015*). We have extrapolated this pattern to other organisms,

**Table 1  Summary of the content of PATACSDB.**

| Feature | Value |
| --- | --- |
| Total number of polyA-carrying transcripts | 197,964 |
| Highest percentage of polyA-carrying transcripts (first 5) | Plasmodium berghei 68.259% |
| | Plasmodium yoelii 17× 64.957% |
| | Plasmodium falciparum 63.539% |
| | Plasmodium chabaudi 63.372% |
| | Plasmodium reichenowi 62.933% |
| Lowest percentage of polyA-carrying transcripts (first 5) | Pythium vexans 0.025% |
| | Saprolegnia diclina vs. 20 0.038% |
| | Leishmania major 0.048% |
| | Phytophthora sojae 0.058% |
| | Salpingoeca rosetta 0.060% |
| Median and average percentage of polyA-carrying transcripts | 2.0% and 7.6% respectively |
| The longest polyA tracks (first 10) | 132 nt—CDO62875 (*Plasmodium reichenowi*) |
| | 131 nt—CDO63348 (*Plasmodium reichenowi*) |
| | 111 nt—ETW31025 (*Plasmodium falciparum fch 4*) |
| | 109 nt—ETW57402 (*Plasmodium falciparum palo alto uganda*) |
| | 107 nt—ETW41820 (*Plasmodium falciparum nf135 5 c10*) |
| | 107 nt—ETW15539 (*Plasmodium falciparum vietnam oak knoll fvo*) |
| | 97 nt—CDO66404 (*Plasmodium reichenowi*) |
| | 95 nt—EUT78604 (*Plasmodium falciparum santa lucia*) |
| | 89 nt—ETW44841 (*Plasmodium falciparum nf135 5 c10*) |
| | 88 nt—ETW48723 (*Plasmodium falciparum malips096 e11*) |

because without further experimental work we have no way to define the minimal polyA pattern in other organisms. We have analyzed eukaryotic Ensembl genomes (*Flicek et al., 2014*) for the presence of this pattern in coding sequences, using only these entries for which coding sequence matched reported translated sequence. This was done not only on standard Ensembl genomes but additional eukaryotic databases like Ensembl Protists and Ensembl Metazoa. As a result, we have identified 197,964 genes in 254 genomes that carry 446,206 polyA tracks.

## PolyA tracks across eukaryotic organisms

In the previous studies (*Koutmou et al., 2015*; *Arthur et al., 2015*) we focused mainly on polyA tracks from human and yeast genomes, using the NCBI (*Pruitt et al., 2014*) database and SGD (*Cherry et al., 1998*) as data sources, respectively. Overall there is a good agreement between our previous analysis and this study for high eukaryotes, while we see some discrepancies for lower eukaryotes such as yeast. For example, in the previous study we have underestimated the number of polyA-carrying genes in yeast by an order of magnitude (29 vs. 369)—a result of different data source.

The percentage of polyA carrying transcripts varies from organism to organism and exceeds 60% for *Plasmodium* species, well known for their AT-rich genome (see Table 1 for summary). However, the distribution of lengths of polyA tracks is quite similar across

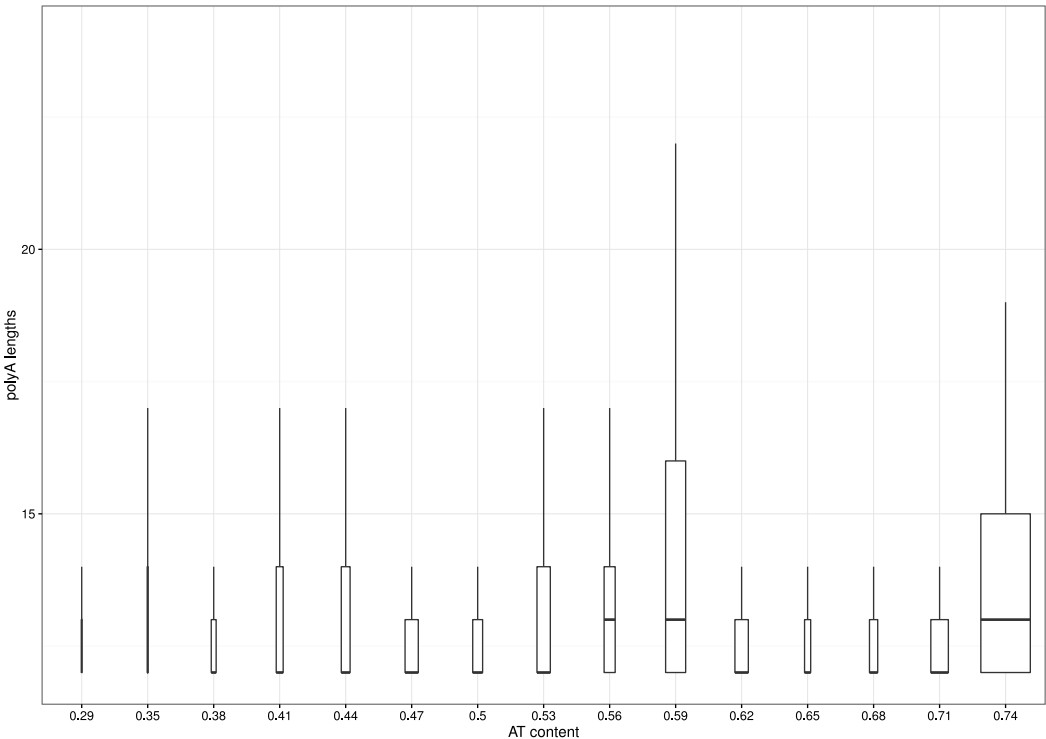

**Figure 1  Distribution of polyA lengths vs. AT-ratio of analyzed genomes.** Data for lengths of polyA were divided into 16 bins distributed evenly across spectrum of AT-richness of analyzed genomes. The width of a box is proportional to the number of observances in a particular bin. Lines denote 1.5*IQR (interquantile range). Outliers were removed for clarity. Data start at length of 12, as this was the length of the minimal pattern used.

whole observed spectrum of AT-content (Fig. 1). It might be that the single *Plasmodium* genus is skewing the distribution, as the species distribution of genomic databases is heavily biased. In humans, only around 1% of transcripts coming from ca. 2% of genes carry polyA track and, as such, are subjects of translational attenuation. This is close to a median across all analyzed genomes. Furthermore, we did not find any correlation between organismal complexity and number of polyA-affected genes. This might indicate that such a feature is a constituent element of translational machinery, unrelated to external factors and regulatory mechanisms.

## Software architecture

The main table consists of protein common name, gene and transcripts Ensembl ids, location of the polyA track expressed as percentage (allows for quick identification of cases where polyA track is either at the end or at the beginning of the protein) and, finally, the identified polyA track with a context of surrounding sequence. All columns are sortable. By default, the table is alphabetically sorted by protein name. Sorting gene and transcript IDs is also alphabetical. Location is sorted numerically. The rows with polyA sequences is sortable by polyA track length, so the user can quickly identify sequences with the longest track in particular organism. Obviously, due to the used pattern, the shortest polyA tracks

have length of 12 nucleotides. To facilitate quick interaction with tables, we have used Bootstrap-table library that allows for easy and intuitive sorting and searching through all fields in particular genome.

The project was created using Python 2.7. To parse biological data, we used Biopython 1.65. To compare protein and cdna sequences, we used local version of NCBI blast+ software v. 2.2.31. To run the web service, we used Flask v.0.10.1. We used SQLite3 database engine and SQLAlchemy for database access. To query the Ensembl database, we used mysql client. We also used two other Python libraries: *xmltodict* and *requests*. The most difficult task was to ensure short pageload times given the large dataset on which we worked. To solve this problem, we have created additional tables in database which contain metadata with the heaviest queries. This solution decreased time of loading more than 20 times.

We have designed a two-step architecture. In the first step, we analyse data from the Ensembl database and create our database with 12A-1 pattern. In the second step, we use created database to provide information to the web service. This architecture allows one to separate obtaining data and running the web service, thus, during analysis of new version of Ensembl data we still can provide data about old version, and changes between versions can be effected in seconds without the notice of the user. In the future, we will work on parallelization process of Ensembl data analysis to speed up the first step. It is likely that polyA segments are not the only sequence determinants of translation efficiency in coding sequences, and further studies will discover more of such motifs or different lengths of minimal polyA pattern for a particular organism. The design of the PATACSDB engine allows for easy modification towards finding and cataloguing of novel sequence patterns.

### Funding
The authors received no funding for this work.

### Competing Interests
The authors declare there are no competing interests.

### Author Contributions
- Malgorzata Habich conceived and designed the experiments, performed the experiments, analyzed the data, contributed reagents/materials/analysis tools, wrote the paper, performed the computation work, reviewed drafts of the paper.
- Sergej Djuranovic conceived and designed the experiments, wrote the paper, reviewed drafts of the paper.
- Pawel Szczesny conceived and designed the experiments, performed the experiments, contributed reagents/materials/analysis tools, wrote the paper, reviewed drafts of the paper.

## Data Availability

https://github.com/habich/PATACSDB/.

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
