# Peer review of "PATACSDB—the database of polyA translational attenuators in coding sequences"

_PeerJ Computer Science, doi:10.7717/peerj-cs.45_

## Round 0.1 · original submission · Major Revisions

The review reports are rather critical. Among others, it is required that the authors must be able to justify the validity of the database content.

·

Basic reporting

The abstract looks concise to me.

I cannot find any figure or table for the manuscript. It would be best if the authors could include some figures and tables for user-friendly explanation. It is important as the manuscript is a database paper.

Experimental design

The experimental setting "pattern 12A allowing for one mismatch" has to be carefully supported on the manuscript. Have the authors tried other experimental setting since this setting can significantly affect the database content ?

Validity of the findings

The sentence "Given that no genomic database reports polyA tracks in coding sequences" may be arguable, please look at the following databases and state your difference to them:
http://exon.umdnj.edu/polya_db/
http://bioinfolab.miamioh.edu/polya/
http://mosas.sysu.edu.cn/utr/
https://www.google.com/webhp?q=polya%20database

Additional comments

It is very nice that the authors have released the scripts which can recreate the
database from the scratch on user's own computer.

The sentence "...is of course a bit simplistic...." is not appropriate on a scientific manuscript.

The first studies
The past studies

gathered researchers' attention
drew researchers' attention

Reviewer 2 ·

Basic reporting

The paper reports generation of a database of coding region polyA sequences from 250 genomes and web service to look this up.
While databases are useful for researchers to look things up in their studies, I am not sure this justifies a scientific report as there is not much experiments or science in this report.
There was also no mention of the validation of the search results at all:should we just trust that the results by the processes reported are reliable? Are all the sequences reported in coding regions? without sequencing errors? I think it is troubling that there is no mentioning of quality examination of the results.

Experimental design

The paper reported the process of collecting the polyA sequences in coding regions. There was no mention of validity check of the results and quality control.

Validity of the findings

No way to judge.

Additional comments

It is essential to validate the results in the database. It is also important to survey the distribution of such sequences in a few model genomes.

---

## Round 0.2 · accepted · Accept

The revised version has addressed most of the relevant comments.